# Food-Grade Polar Extracts from Sea Fennel (*Crithmum maritimum* L.) By-Products: Unlocking Potential for the Food Industry

**DOI:** 10.3390/foods14132304

**Published:** 2025-06-28

**Authors:** Aizhan Ashim, Lama Ismaiel, Benedetta Fanesi, Ancuta Nartea, Antonietta Maoloni, Oghenetega Lois Orhotohwo, Helen Stephanie Ofei Darko, Paolo Lucci, Lucia Aquilanti, Deborah Pacetti, Roberta Pino, Rosa Tundis, Monica Rosa Loizzo

**Affiliations:** 1Department of Agricultural, Food and Environmental Sciences, Università Politecnica delle Marche, 60131 Ancona, Italy; aizhan.ashim.h@gmail.com (A.A.); l.ismaiel@univpm.it (L.I.); b.fanesi@univpm.it (B.F.); a.nartea@univpm.it (A.N.); a.maoloni@univpm.it (A.M.); o.orhotohwo@pm.univpm.it (O.L.O.); h.s.o.darko@pm.univpm.it (H.S.O.D.); l.aquilanti@univpm.it (L.A.); 2Department of Pharmacy, Health and Nutritional Sciences, University of Calabria, 87036 Rende, Italy; roberta.pino@unical.it (R.P.); rosa.tundis@unical.it (R.T.); monica_rosa.loizzo@unical.it (M.R.L.)

**Keywords:** halophyte, Apiaceae, antioxidant, antimicrobial, polyphenols, carotenoids, tocopherols, TPC, TFC

## Abstract

*Crithmum maritimum* L. is a halophyte with antioxidant and antimicrobial potential for the food industry. Pruning generates a by-product composed of woody stems, old leaves, and flowers. To valorize this underutilized and largely unexplored biomass, food-grade polar extraction (hydroethanolic vs. aqueous) was applied. The extracts were characterized for their bioactive compounds (polyphenols, tocopherols, carotenoids, total phenols (TPC) and total flavonoids (TFC)). Further, the extracts were assessed for their in vitro antioxidant activity (ABTS, DPPH, FRAP, and β-carotene bleaching) and antimicrobial activity against eight target strains ascribed to *Escherichia coli*, *Staphylococcus aureus*, and *Listeria innocua*. The hydroethanolic extract exhibited higher concentration of bioactives compared to the water extract and raw by-product. The β-carotene bleaching test revealed that both extracts are potent inhibitors of lipid peroxidation. The aqueous extract showed no antimicrobial activity, while the ethanolic extract exhibited strain-dependent behavior against *S. aureus* and *L. innocua* but not *E. coli*. The minimum inhibitory concentration and the minimum bactericidal concentration of the ethanolic extract against *S. aureus* were 2.5 MIC and 10.0 MBC mg/mL, respectively. Ethanolic extracts could potentially be used in food formulations to enhance lipid peroxidation resistance and antimicrobial capacity as food-grade natural preservatives.

## 1. Introduction

*Crithmum maritimum* L., commonly known as sea fennel or rock samphire, is a halophyte plant belonging to the Apiaceae family. The natural habitat of *C. maritimum* includes rocks, sands or cliffs along seacoasts, especially in the Mediterranean region, where it can tolerate saline and low-moisture soils. From Shakespeare’s tales to the adventures of Moby Dick, the tender leaves of *C. maritimum* have become the subject of numerous studies, meeting both the taste preferences of Mediterranean consumers and the need for sustainability practices [1]. Pickles, preserves, and condiments made from sea fennel leaves are featured in the traditional cuisines of Italy, Greece, Croatia, Turkey, Portugal, and Spain [2,3]. To date, several innovative products, such as pesto-like sauces and fermented sea fennel, have been developed at the laboratory or pilot scale [4]. Small- to medium-sized producers are now developing sea fennel crops across Mediterranean countries, producing vegetable biomass such as leaves and field residuals [3]. Cultivation does not require irrigation or agrochemicals, and the climate change score (6.81 × 10^−2^ kg CO_2_ eq/kg of fresh sea fennel) is better than other Apiaceae (i.e., celery and carrots), highlighting its superior environmental performance [5,6]. Sea fennel cultivation follows a seasonal cycle, where seedlings are raised in a greenhouse and then transplanted to open fields. Planting typically occurs between October and November, while hand harvesting of mature leaves takes place in the second year, from May to July. The plants then regenerate, producing new leaves until reaching full flowering by August. During the cold winter months, the plants are cut at ground level to ensure healthy growth for the following season. The cutting phase produces a by-product composed of underutilized woody parts, stems, old leaves, and flowers [6]. It has been reported that, on average, the waste portion of the sea fennel plant is about 40%, and a plant yields 30–50 g of edible leaves [7,8].

If *C. maritimum* nurseries are replicated across Mediterranean areas, the biomass of *C. maritimum* by-products could reach high volumes. Currently, the only use of *C. maritimum* by-products seems to be in the pharmaceutical sector, according to an Italian company (Rinci s.r.l.), although there is no official documentation regarding this. However, it is reasonable to expect that the field by-product could serve as a source of bioactive compounds, as already demonstrated for the leaves and flowers of *C. maritimum* [9], and their residual waste after essential oil extraction [10]. Phenolic acids, flavonoids, carotenoids such as lutein and β-carotene, and tocopherols were recently reported in high concentrations in wild leaves, flowers, and one cultivated sample of *C. maritimum* [11].

Bioactive compounds, particularly essential oils and chlorogenic acids, in sea fennel are responsible for its functional properties, such as antioxidant and antimicrobial activities [12]. Apolar extracts have shown good antimicrobial activity against fungi, pathogens, and spoilage microorganisms. Petroleum ether and acetone extracts exhibited greater inhibitory effects on the growth of *Staphylococcus aureus* than hydroalcoholic extracts [13]. Chloroform extracted falcarindiol, a potential antibacterial agent against *Pseudomonas* strains [14]. Avoiding the use of organic solvents, water and hydroethanolic extracts are a greener option; however, little is known about the antimicrobial activity provided by the underutilized residual biomass of sea fennel.

Literature analysis reinforces the potential use of sea fennel as a natural preservative, but further investigation is needed to identify the microbial strains sensitive to sea fennel extracts [13,15]. For instance, the antibacterial activity of different extracts varied depending on the strain tested. Sea fennel stems exhibited the strongest activity against the Gram-negative bacterium *Pseudomonas aeruginosa* [8,16]. The antimicrobial activity of essential oils is better documented in the literature than that of polar extracts against some Gram-positive and Gram-negative spoilage bacteria [12].

Given the significant potential of sea fennel extracts and essential oils as antioxidants and antimicrobials, and the lack of studies on their application as natural food preservatives, further research is required for their optimal use in this emerging field. Sea fennel has the potential to provide the food industry with antioxidants and antimicrobials that could be used to extend the shelf life of perishable foods [17,18,19]. Indeed, scientific evidence and consumer preference for healthy and natural foods are driving the food industry towards the substitution of synthetic additives with natural preservatives such as essential oils, extracts, or selected bioactive compounds [20,21]. These molecules could be integrated into food formulations [22] or added in controlled atmospheres, films, and coatings [23], meeting the “clean label” market demands [24]. Thus, sea fennel polar extracts enriched in phenolic compounds could be integrated into food packaging materials to develop films, multilayers, and coatings with enhanced functional properties, as reviewed by Orhotohwo et al. [8].

In this framework, this study aimed to valorize the underutilized residual biomass of *C. maritimum* crop after its cultivation as a potential ingredient in food preservation, in the form of an extract. To this end, for the first time, the *C. maritimum* by-product was chemically characterized and extracted with affordable, eco-friendly water and ethanol using ultrasound technology. Extracts were assessed for their polyphenols profile, bioactive compounds content, antioxidant and antimicrobial activities against strains of foodborne pathogens (*Escherichia coli* and *Staphylococcus aureus*) and a surrogate (*Listeria innocua*) for pathogenic *Listeria monocytogenes*.

## 2. Materials and Methods

### 2.1. Chemicals

Acetonitrile, methanol, ethanol, isopropanol, hexane, ethyl acetate, water, formic acid and chlorogenic acid were purchased from Sigma-Aldrich. All other solvents and reagents were of analytical grade; standards (>99% pure), including lutein, α- and γ-tocopherol, were purchased from Sigma-Aldrich (St. Louis, MO, USA).

### 2.2. Sea Fennel By-Product Supply

*C. maritimum* by-product was kindly provided by a local company, Rinci s.r.l. (Castelfidardo, Italy; 43°26′51″ N 13°33′21″ E). The by-product consisted of heterogeneous materials such as wood parts, stems, old leaves, and flowers of cultivated plants. After a visual inspection to ensure the absence of mold, the biomass was stabilized by air-drying in a De Cloet Dryer at 40 °C until it reached a relative humidity (RH) of less than 15% at the company facilities. The dried by-product was sealed in hermetic bags, transferred to the laboratory and stored at room temperature in the dark. The hardest inedible parts were manually removed and, then, the by-product was ground with a cutter mill using a 6 mm grid to obtain a fine powder (BP-RAW) prior to extract preparation.

### 2.3. Preparation of Aqueous and Hydroethanolic Extract from Sea Fennel By-Product

The aqueous extract (BP-W) was prepared using BP-RAW and deionized water at a ratio of 1:20 *w*/*v*, as reported by Alemán et al. [25]. Briefly, 10 g of fine powder BP-RAW was mixed with 200 mL of deionized water and stirred for 18 h at room temperature in the dark, then centrifuged (3500 rpm, 5 min), filtered through a 0.45 µm regenerated cellulose filter, freeze-dried, and stored at −20 °C for later analysis.

The hydroethanolic extract (BP-ET) was obtained from BP-RAW (10 g) using an ethanol/water mixture (80:20, *v*/*v*) at a solid to solvent ratio of 1:20 *w*/*v*. The mixture was subjected to ultrasound-assisted extraction for 30 min, then centrifuged (3500 rpm, 5 min) and filtered through a 0.45 µm regenerated cellulose filter. Ethanol was evaporated using a vacuum rotary evaporator at 40 °C, and the remaining water was removed by freeze-drying. The extraction was performed in triplicate for both the aqueous and hydroethanolic extracts and the yields were calculated and expressed as %. All extracts were stored at −20 °C until further analysis. Figure 1 illustrates sea fennel preparation steps for BP-RAW, BP-W and BP-ET.

Equation (1) was used to determine the extraction yield of BP-ET and BP-W extracts:(1)$$\mathrm{Extraction}\;\mathrm{yield}\;(\%)=\frac{W_{1}}{W_{2}}\times 100$$
where W_1_ is the mass of the final dried extract and W_2_ is the mass of the initial sample prior to extraction.

### 2.4. Total Phenols and Flavonoids Contents

BP-RAW and the two freeze-dried extracts (BP-W and BP-ET) were analyzed for both total phenolic content (TPC) and total flavonoid content (TFC). The extraction was carried out according to Fanesi et al. [26] with minor modifications. Briefly, 0.25 g of dry sample was extracted with 3.75 mL ethanol/water (50:50, *v*/*v*) at 40 °C for 1 h under agitation, then centrifuged (3500 rpm, 10 min at 4 °C). The supernatant was collected and subjected to TPC, TFC, as well as polyphenol profile analysis. TPC was determined according to the Folin–Ciocalteu method. Therefore, 20 µL of the supernatant was added to 1.58 mL water and 100 µL of Folin reagent, and left for 7 min in the darkness; then, 300 µL of sodium carbonate solution was added. After 30 min, the absorbance was measured at λ = 750 nm using a UV-Vis spectrophotometer (UV-31 SCAN, Onda, Beijing, China), and results were expressed as mg gallic acid equivalents (GAE) per g of dry sample for literature comparison purposes. For TFC quantification, the sample was mixed with 2% aluminum chloride solution and allowed to react at room temperature for 15 min [27]. The absorbance was then measured at λ = 430 nm, using the same UV-Vis Agilent 8453 spectrophotometer. The TFC was expressed as mg quercetin equivalents (QE)/g dw (dry weight).

### 2.5. Polyphenol Profile

Polyphenol profile was determined by injecting 2 μL of each sample into an Acquity UHPLC H-Class system (Waters Corporation, Milford, CT, USA) equipped with a photodiode array (PDA) detector and mass spectrometer (MS). For the analysis, an Accucore C18 column (150 × 2.1 mm, 2.6 μm; ThermoScientific, Waltham, MA, USA) was used, and the temperature was set at 40 °C. The mobile phases consisted of water (eluent A) and acetonitrile (eluent B), both acidified at 0.1% with formic acid. The gradient started at 5% B for 1 min, raised to 70% B at 20 min and to 95% B at 25 min, then kept in isocratic for 5 min before restoring the initial conditions. The flow was set at 0.4 mL/min. The PDA spectra were acquired in full scan from 200 to 500 nm. MS was operated in negative electrospray ionization from 100 to 610 *m*/*z*. The cone voltage was 15 V, and the capillary voltage was 0.8 kV. Phenolic compounds were tentatively identified based on retention time, absorbance spectra by PDA and mass-to-charge ratio (*m*/*z*) spectra obtained in negative ionization mode for each compound, compared with PubChem databases. Quantification was performed using the peak area of compounds extracted from PDA at 280 nm and an external calibration curve based on chlorogenic acid in the range of 10–500 ng/μL (R^2^ = 0.9987), and results were expressed as mg of chlorogenic acid equivalent (CGA) eq/g dw.

### 2.6. Carotenoids and Tocopherols

Carotenoids and tocopherols were simultaneously analyzed in BP-RAW, BP-W, and BP-ET, as reported by Stinco et al. [28] with minor modifications. Briefly, 0.1 g of dry sample was mixed with 10 mL of MilliQ water, vortexed and centrifuged at 6000× *g* for 5 min to remove the aqueous part. Later, 10 mL of hexane/acetone (1:1, *v*/*v*) was added, vortexed and centrifuged using the same conditions as before. The colored fraction was recovered in a balloon, and the extraction was repeated using 6 mL of fresh solvent (hexane/acetone, 1:1, *v*/*v*) until color depletion. The pooled fractions were taken to dryness using a rotary evaporator (Rotavapor R-210, Buchi, Switzerland), and the residue was then resuspended in 1.5 mL isopropanol prior to injection. An Aquity UHPLC H-Class system (Waters Corporation, Milford, MA, USA) equipped with PDA and Accucore C18 column (150 × 2.1 mm, 2.6 μm; ThermoScientific) was used to perform chromatographic separation of analytes. The column temperature was set at 28 °C, while samples were kept at 10 °C. The mobile phases consisted of acetonitrile (A), methanol (B), and ethyl acetate (C). The gradient (A:B:C) started at 85:15:0; at 5 min, 60:20:20, maintained for 2 min; at 9 min, 85:15:0 until 12 min. The UV-visible spectra were obtained between 200 and 500 nm, and the chromatograms were processed at 450 nm for carotenoids and 285 nm for tocopherols. Retention times and absorbance spectra of pure standards were used to identify the carotenoids and tocopherols. Their quantification was performed via external calibration injected in a range of 1–100 μg/mL for lutein and of 5–100 μg/mL for *α*- and γ- tocopherol, obtaining good correlation coefficients (R^2^ > 0.9988).

### 2.7. Antioxidant Activity

The radical scavenging activity was investigated using two different procedures: 2,2′-azino-bis (3-ethylbenzothiazoline-6-sulfonic acid) (ABTS) and 1,1-diphenyl-2-picrylhydrazyl (DPPH) radical scavenging assays. Briefly, for the ABTS test, the ABTS•^+^ radical cation solution was prepared by mixing ABTS (2 mM) with K_2_S_2_O_8_ (70 mM). After 24 h in the dark at room temperature, 1 mL of the solution was diluted with ethanol to reach an absorbance of 0.70 at λ = 734 nm. The solution obtained was added to the extract at different concentrations (1–400 μg/mL) and left to react for 6 min at room temperature. The absorbance was then measured at λ = 630 nm, using an Agilent 8453 UV-Vis spectrophotometer (Agilent Technologies, Milan, Italy) [29].

For the DPPH test, a solution of DPPH (1 × 10^−4^ M in MeOH) and the sample at different concentrations (1–1000 μg/mL) were mixed. After 30 min of incubation at room temperature in the dark, the absorbance was measured at λ = 490 nm, using an Agilent 8453 UV-Vis spectrophotometer (Agilent Technologies, Milan, Italy).

The β-carotene bleaching test (coupled oxidation of β-carotene and linoleic acid) estimates the relative ability of sea fennel by-products to scavenge the linoleic acid peroxide radical (LOO•) that oxidizes β-carotene in the emulsion phase. Briefly, 1 mL of β-carotene (0.2 mg/mL in chloroform) was mixed with linoleic acid (20 μL) and Tween 20 (200 μL). After evaporation of chloroform and dilution with water, 5 mL of the obtained emulsion was collected and transferred into microplate wells containing the sample (200 μL) at different concentrations (2.5–100 μg/mL). After 30 and 60 min of incubation at 45 °C, the absorbance was measured at λ = 470 nm [29].

In all tests, data were expressed as IC_50_ values. Ascorbic acid was used as positive control in the DPPH and ABTS tests, whereas propyl gallate was used as the positive control in the β-carotene bleaching test.

For the FRAP test, a solution of FRAP reagent, tripyridyltriazine (TPTZ, 10 mM), acetate buffer (300 mM), FeCl_3_ (20 mM), and HCl (40 mM) was prepared. The sample, at a concentration of 2.5 mg/mL, was then mixed with distilled water (900 μL) and FRAP solution (2 mL). After 30 min of incubation at room temperature in the dark, the absorbance was measured at λ = 593 nm, using the same UV-Vis Agilent 8453 spectrophotometer. In this test, BHT was used as the positive control [29].

### 2.8. Bacterial Strains and Antimicrobial Susceptibility Testing

The antimicrobial activity of the extracts was assayed against *Escherichia coli* (ATCC 25922, Ec 2), *Staphylococcus aureus* (ATCC 29213, ATCC 25923, DSM 20231), and *Listeria innocua* (Li 1, Li 2, Li 3). The strains *E. coli* ATCC 25922, *S. aureus* ATCC 29213, and *S. aureus* ATCC 25923 were purchased from the American Type Culture Collection (ATCC), Manassas, VA, USA [30,31], while *S. aureus* DSM 20231 was obtained from the Deutsche Sammlung von Mikroorganismen und Zellkulturen (DSMZ) [31]. The *E. coli* strain Ec 2 was kindly provided by the Istituto Zooprofilattico Sperimentale dell’Umbria e delle Marche, Perugia (Italy), while all *L. innocua* strains were kindly provided by the Department of Agricultural, Forest and Food Sciences (DISAFA), University of Turin, Italy. All the strains were maintained at −80 °C in a mixture of glycerol and Brain Heart Infusion (BHI) broth (Liofilchem S.r.l., Roseto degli Abruzzi, Italy), at a 2:3 (*v*/*v*) ratio. Prior to use, *E. coli*. and *S. aureus* strains were subcultured in Mueller-Hinton (MH) broth (Merck KGaA, Darmstadt, Germany), while *L. innocua* strains were subcultured in Tryptic Soy Broth (TSB) (VWR, Radnor, PA, USA); all strains were incubated at 37 °C for 24 h.

Prior to antimicrobial susceptibility testing, the aqueous freeze-dried extract was resuspended at a concentration of 400 mg/mL in distilled water and sterilized through filtration (0.22 µm syringe filter, Starlab International GmbH, Hamburg, Germany), while the ethanol freeze-dried extract was resuspended at a concentration of 200 mg/mL in dimethyl sulfoxide (DMSO) (Merck KGaA).

#### 2.8.1. Agar Well Diffusion Method

The antimicrobial activity was evaluated using the agar well diffusion method, as described by Balouiri et al. [32], with slight modifications. All strains were subcultured overnight in MH broth (*E. coli* and *S. aureus*) or TSB (*L. innocua*), and their density was adjusted to a turbidity of 0.5 McFarland using a Shimadzu UV-1800 spectrophotometer (Shimadzu Corporation, Kyoto, Japan) in a 0.85% sterile NaCl solution. A sterile cotton swab was dipped into each suspension and used to inoculate MH agar (Merck) (*E. coli* and *S. aureus*) or Tryptic Soy Agar (TSA) (Liofilchem) (*L. innocua*) plates, which were then perforated in the center to create wells with a diameter of 7 mm. Each well was filled with 50 µL of the water extract resuspended in water (400 mg/mL) or the ethanol extract resuspended in DMSO (200 mg/mL). For the ethanol extract antimicrobial activity determination, plates filled with 50 µL of pure DMSO were used as negative controls. The plates were incubated at 37 °C for 18 ± 2 h, and the inhibition zones (mm) were measured. Analyses were performed in duplicate, and the results were expressed as mean values ± standard deviation.

#### 2.8.2. Broth Microdilution Method

The broth microdilution method was performed according to EUCAST guidelines [33,34] with slight modifications. Briefly, MH broth (Merck) was used for *E. coli* and *S. aureus* strains, and TSB (VWR) was used for *L. innocua* strains. Extract concentrations ranging from 100 to 0.1 mg/mL for the water extract and from 10 to 0.01 mg/mL for the ethanol extract were tested. For the ethanol extract assay, microtitration plates containing pure DMSO concentrations ranging from 5% to 0.005% were used as negative controls. Each strain was inoculated at a load of 5 × 10^5^ CFU/mL, and the inoculum was verified by sampling 10 µL from the growth control well immediately after inoculation, diluting it in 10 mL of 0.85% sterile NaCl solution and plating 100 µL of the resulting suspension onto MH agar (Merck) for *E. coli* and *S. aureus* strains, or TSA (Liofilchem) for *L. innocua* strains. The microtitration plates were incubated at 37 °C for 18 ± 2 h. The minimum inhibitory concentration (MIC), defined as the lowest concentration of the extract that completely inhibits visible growth, was determined for the water extract only, as MIC evaluation for the ethanol extract was prone to error due to the turbidity of the extract in the culture medium. The minimum bactericidal concentration (MBC) was determined by sampling 10 µL from each well and plating it on the appropriate culture medium. The plates were incubated at 37 °C for 24 h, and MBC was defined as the lowest concentration of antimicrobial agent that killed 99.9% of the inoculum [32]. Analyses were performed in duplicate, and the results were expressed as mean values ± standard deviation.

#### 2.8.3. Agar Dilution Method

The MIC for the ethanol extract was determined through the agar dilution method, as described by EUCAST [35] with slight modifications. Briefly, a dilution series of the extract was prepared in 15 mL sterilized centrifuge tubes with a final volume of 0.5 mL, to which 9.5 mL of the appropriate molten agar (MH agar for *E. coli* and *S. aureus*, or TSA for *L. innocua*) was added. The mixture was thoroughly mixed and poured into 60 mm Petri dishes. Extract concentrations ranging from 2.5 to 10 mg/mL were tested, with a control plate containing no extract. Additional negative control plates containing DMSO concentrations ranging from 1.25% to 5% were also prepared, along with one plate containing no DMSO.

All strains were subcultured overnight in MH broth (*E. coli* and *S. aureus*) or TSB (*L. innocua*), and their density was adjusted to approximately 8 Log CFU/mL using a Shimadzu UV-1800 spectrophotometer (Shimadzu), then further diluted to 7 Log CFU/mL in the appropriate media. Spots containing 4 Log CFU/mL of each strain were inoculated onto the agar plates by applying 1 µL of the microbial suspension. The plates were incubated at 37 °C for 18 h, and the MIC was determined as the lowest concentration of the extract that completely inhibited visible growth on the surface of the plates. As this is an exploratory study to evaluate the intrinsic antimicrobial activity of a natural extract, only negative controls were employed. Analyses were performed in duplicate, and the results were expressed as mean values ± standard deviation.

### 2.9. Statistical Analysis

One-way analysis of variance (ANOVA) was performed through the Tukey–Kramer honest significant difference (HSD) test (*p* ≤ 0.05) using JMP Version 11.0.0 software (SAS Institute Inc., Cary, NC, USA).

## 3. Results and Discussion

### 3.1. Polyphenols Profile and Bioactive Compounds Content

The characterization of BP-RAW, BP-W and BP-ET in terms of their polyphenol composition, tocopherols, carotenoids, TPC and TFC is presented in Table 1.

Both BP-W and BP-ET extracts exhibited significantly higher concentrations of hydroxycinnamic acids compared to BP-RAW. The latter contained only 4.31 mg/g of hydroxycinnamic acids, whereas BP-W and BP-ET showed approximately five-fold higher levels (22.40 mg/g and 22.99 mg/g, respectively). Within this class of compounds, caffeoylquinic acids, mainly represented by chlorogenic acid, accounted for ~ 61.3% of the total polyphenols, followed by dicaffeoylquinic acids, coumaroylquinic acids, feruloylquinic acids, and caffeic acid (Figure 2). Chlorogenic acid has been reported by several authors as the predominant constituent in hydroethanolic extracts of sea fennel [15,36,37]. The antioxidant properties of various chlorogenic acid isomers were studied by Xu et al. [38], who found that dicaffeoylquinic acids exhibited superior antioxidative properties compared to caffeoylquinic acids. Generally, it has been demonstrated that the presence of ortho-hydroxyl groups on the aromatic ring enhances the antioxidant activity, as does the number of hydroxyl groups. On the other hand, BP-W showed a higher extraction yield (17.42% ± 2.36) than BP-ET (14.05% ± 2.58). This suggests that, although the aqueous method produced a greater overall yield, the hydroethanolic extraction was more successful in isolating bioactive compounds.

BP-ET had the highest concentration of flavonoids (1.69 mg/g), followed by BP-W (0.77 mg/g) and BP-RAW (0.71 mg/g). Specifically, the quercetin-glucoside isomer (*m*/*z* 475), isoquercetin (*m*/*z* 463), diosmin (*m*/*z* 607) and apigenin hexoside (*m*/*z* 431) were tentatively identified in all samples, considering that sea fennel’s antioxidant activity may be significantly influenced by the flavonoid content [39].

TPC and TFC confirmed the efficiency of the hydroethanolic extraction as it resulted in a total polyphenol content of 55.10 mg GAE/g DW, which was approximately twice that of BP-RAW (27.93 mg GAE/g DW) and higher than that of the aqueous extract (49.27 mg GAE/g DW). The same trend was recorded for TFC, and BP-RAW had the lowest flavonoid concentration (2.31 mg QE/g DW), whereas BP-ET exhibited a total of 9.62 mg QE/g DW, which is double that of the aqueous extract (5.27 mg QE/g DW). The literature reveals significant variability in the TPC and TFC values of *C. maritimum* due to factors, such as collection area, period of collection, plant organs used, and extraction methods. For example, leaves of sea fennel from France contained 26 mg GAE/g DW for TPC and 12 mg QE/g DW for TFC [40]. In contrast, leaves from plants collected in Greece exhibited TPC values ranging from 2.55 to 10.84 mg GAE/g DW and TFC values ranging from 2.25 to 15.08 mg CE/g DW [41]. Other TPC concentrations (23–33 mg GAE/g DW) were reported in relation to the period of collection, indicating temporal variation in the phenolic content of sea fennel [15].

Generally, the results obtained from the constituent analysis of sea fennel by-product may align closely with their corresponding ones in the flowers and leaves [11]. BP-RAW contains significant amounts of lutein (29.62 mg/kg DW), α-tocopherol (18.98 mg/kg DW) and γ-tocopherol (19.23 mg/kg DW) compared to the extracts. These values are comparable to those reported for wild sea fennel flowers, whereas only the γ-tocopherol content aligns with that found in wild sea fennel leaves [11]. The tocopherol content of sea fennel flowers from three distinct populations varied significantly, according to a study from along the Croatian Adriatic coast. α-tocopherol was most prevalent in one sample (73.62 mg/kg DW), whereas its concentration was more than 29-times lower in the other two. These results imply that sea fennel’s tocopherol composition is highly influenced by the harvest’s geographic origin [42]. Despite the compositional similarities to flowers, the levels of these compounds in the by-product are generally lower than those reported in sea fennel leaves, especially for lutein concentrations (102.54 to 190.89 mg/kg DW) and α-tocopherol (194.29 to 598.12 mg/kg DW) and, to a lesser extent, in the case of γ-tocopherol concentrations (23.65 to 26.20 mg/kg DW) [11]. BP-ET revealed notably higher amounts of lutein (56.41 mg/kg DW) and α-tocopherol (30.89 mg/kg DW), while they were not detected in BP-W.

The phytochemical profile of *C. maritimum* can vary depending on many factors (i.e., habitat, plant organ, vegetation period), but standardization through cultivation can mitigate the differences [11,43], and this is an advantage of the *C. maritimum* by-product, even if, in general, lower phytochemicals are present in the by-product with respect to its edible leaves.

### 3.2. Antioxidant Activity

Table 2 exhibits the antioxidant activity of aqueous and hydroethanolic extracts of sea fennel. The sea fennel by-product water extract (BP-W) demonstrated more promising radical scavenging activity, with half-maximal inhibitory concentration (IC_50_) values of 119.91 μg/mL for ABTS and 120.75 μg/mL for DPPH. The hydroethanolic extract (BP-ET) showed similar potency against the ABTS^+^ radical cation, with an IC_50_ value of 121.82 μg/mL. Both extracts were also effective at inhibiting lipid peroxidation, as indicated by the β-carotene bleaching test. Specifically, BP-ET showed IC_50_ values of 3.38 μg/mL and 3.27 μg/mL after 30 and 60 min of incubation, respectively. The FRAP value of BP-ET was close to that of the positive control, butylated hydroxytoluene (BHT), with values of 63.19 μM Fe(II)/g compared to 60.20 μM Fe(II)/g for BHT at 2.5 mg/mL.

Previous studies have shown that sea fennel by-products from Tunisia exhibited lower radical scavenging potential, with IC50 values for DPPH of 726 μg/mL for stems, 706 μg/mL for leaves, and 500 μg/mL for flowers [16]. Alemán et al. (2019) also reported that the antioxidant activity, measured using the ABTS and FRAP methods, was 1.4-fold and 2.1-fold higher, respectively, in the hydroethanolic extract compared to the aqueous extract [25]. Notably, the antioxidant activity of the BP-W and BP-ET extracts in this study differed significantly, with the ethanol extract showing values 2.9-times higher in DPPH and 1.3-times higher in FRAP compared to the aqueous extract.

It is important to note the significant variability in the antioxidant potential of sea fennel extracts, which could be influenced by factors such as the plant organ used, the reproductive stage or period of collection, and the extraction procedure applied. Jallali et al. (2012) highlighted that extracts from plants harvested in summer are richer in bioactive compounds, which likely contributes to higher antioxidant activity [44]. The observed high IC_50_ value in the DPPH assay for BP-ET may be due to the differential solubility and accessibility of antioxidant compounds in the assay medium. The ABTS assay, which can be performed in both aqueous and organic solvents, may allow for better interaction of both polar and non-polar antioxidants with the ABTS radical cation. This could partially explain why no statistically significant differences were observed between BP-ET and BP-W in the ABTS results. Additionally, matrix effects and the potential presence of antagonistic compounds in the ethanol extract (e.g., polysaccharides, proteins, or other interfering substances) could also influence the effective antioxidant response in the DPPH assay. A positive correlation was observed between the total phenolic content (TPC) and antioxidant activity, with a quite strong one found for FRAP and DPPH assays (r = 0.7). A comparison of different extraction methods reveals that microwave-assisted extraction (MAE) yields better results than conventional maceration, as it produces extracts richer in phenolic compounds. Conversely, Soxhlet extraction results in extracts with higher radical scavenging activity and reducing power properties [11,44].

Interestingly, except for the β-carotene bleaching test, a positive correlation was observed for TPC and TFC (r = 1). This suggests that TPC and TFC may be the primary contributors to the antioxidant activity in *C. maritimum* extracts, as their contents were positively associated with antioxidant activity. Lemoine et al. assessed the TPC of crude *C. maritimum* leaf extract (1:2 water/ethanol), which was found to be 33.3 mg CGA/g DW. Upon fractionating the extract with 20%, 40%, and 60% methanol, the authors found that these fractions contained 3.4-, 5-, and 2.3-times more phenolic compounds, respectively, than the crude extract [45]. The authors recommended using the crude polar extract for its antioxidant properties, while methanol fractionation at different concentrations could be explored for additional health benefits, such as anti-diabetic, anti-obesity, anti-inflammatory, and anti-aging properties.

### 3.3. Antimicrobial Activity

Ethanol, as one of the solvents used for extraction in this study, is considered compliant with good manufacturing practices (GMPs) and poses no significant risk to human health. Furthermore, the selection of these solvents adheres fully to European Union Directive 2009/32/EC [46], which takes into account considerations of toxicity, environmental impact, and the efficiency of bioactive substance extraction.

The refuse parts of the new sea fennel crops were extracted using food-grade hydro-alcoholic (water/ethanol) solvent mixtures, employing green extraction technologies such as ultrasound. These methods are eco-friendly and efficient, ensuring high-quality extracts while minimizing the environmental footprint.

The results of the antimicrobial activity tests are presented in Figure 3, Figure 4, Figure 5 and Figure 6 and Table 3. Interestingly, the water extract did not produce any inhibition zones against any of the tested strains (Figure 3), and the minimum inhibitory concentration (MIC) values for the water extract were found to be >100 mg/mL against all the strains tested. This suggests that the water extract alone may not possess significant antimicrobial activity under the conditions tested. In more detail, both the low concentration of phenolic compounds (TPC and TFC values) and the composition of the extract itself may have determined the absence of the antimicrobial activity of the water extract. Indeed, different solvents have the capability of extracting different molecules [47], within phenolic compounds, displaying a diverse antimicrobial activity towards selected pathogens [48].

The ethanol extract demonstrated notable antimicrobial activity, with the largest inhibition growth zone observed for *Staphylococcus aureus* DSM 20231 (4.50 ± 0.00 mm). Additionally, the ethanol extract exhibited comparable inhibition zones against other *S. aureus* strains, including *S. aureus* ATCC 29213 (3.31 ± 0.09 mm) and *S. aureus* ATCC 25923 (3.13 ± 0.00 mm) (Figure 2, Table 3). For the three strains of *L. innocua*, the inhibition zones were also significant, with values of 3.00 ± 0.18 mm for Li 1, 3.13 ± 0.00 mm for Li 2, and 3.19 ± 0.27 mm for Li 3 (Figure 3, Table 3).

These findings highlight the effectiveness of the ethanol extract in inhibiting the growth of a range of bacterial strains, particularly *S. aureus* and *L. innocua*, suggesting its potential as a natural antimicrobial agent.

The findings by Houta et al. on the antibacterial properties of methanolic extracts from various *C. maritimum* parts, specifically flowers, stems, and leaves, highlight stronger antimicrobial activity compared to the current study. Houta et al. reported inhibition zones of 11 mm, 10 mm, and 10 mm for the flower, stem, and leaf extracts, respectively, against *S. aureus* ATCC 25923 and 10 mm, 11 mm, and 10 mm for *Escherichia coli* ATCC 29212 [16]. In contrast, the current study found much smaller inhibition zones of 3.31 ± 0.09 mm (*S. aureus* ATCC 29213) and 3.13 ± 0.00 mm (*S. aureus* ATCC 25923) for the ethanol extract, indicating significantly lower antimicrobial activity.

These differences can be attributed to the varying extraction solvents used. Houta et al. employed a methanolic extract (5 g of dry powder in 50 mL of methanol), which likely resulted in a higher yield of bioactive compounds, leading to more pronounced antibacterial effects. In contrast, the present study utilized higher concentrations of the ethanol extract, resulting in lower inhibition zone sizes.

The comparatively higher MIC and MBC values obtained in our study may be linked to the complex and heterogeneous nature of the sea fennel by-product matrix, which includes woody stems, old leaves, and flowers, unlike extracts derived solely from leaves or flowers. Supporting this variability, Al Rugaie et al. reported that hydroethanolic extracts from three halophyte species exhibited a broad spectrum of antimicrobial activity against various microorganisms, including different strains of *S. aureus* ATCC and *B. cereus* ATCC. Specifically, *Euphorbia chamaesyce* showed MIC values ranging from 12.5 to 25 mg/mL and MBC values from 25 to 50 mg/mL; *Haloxylon salicornicum* exhibited MIC values of 0.78 to 12.5 mg/mL and MBC values of 1.56 to 25 mg/mL; while *Bassia arabica* displayed MIC values between 12.5 and 25 mg/mL and MBC values from 25 to 50 mg/mL [49]. These findings underscore that the antimicrobial efficacy of *Crithmum maritimum* extracts is significantly influenced by key variables such as the extraction solvent, extract concentration, and the specific plant part utilized. The comparatively smaller inhibition zones observed in our study may be attributed to the heterogeneous composition of the by-product material, which includes a mix of woody stems, old leaves, and flowers. This mixed tissue origin should be recognized as a potential limiting factor in bioactivity. Overall, these observations highlight the importance of standardizing experimental conditions, particularly in terms of plant material selection and extraction methodology, to enable more accurate comparisons across studies.

According to Correia et al. [13], extracts of sea fennel flowers and schizocarps made with petroleum ether and acetone showed stronger inhibitory effects on *S. aureus* growth. In contrast, a study by Souid et al. [16] found that sea fennel leaves had significant antimicrobial activity against *S. aureus* in a hydroethanolic extract. Both studies suggest that the high concentration of chlorogenic and neochlorogenic acids in *C. maritimum* extracts may contribute to their antimicrobial activity, as both isomers have been shown to cause bacterial membrane disruption, alter intracellular potential, and ultimately lead to bacterial death. This antibacterial action of chlorogenic acid derivatives has been documented against a wide range of bacteria, such as *S. aureus*, *S. pneumoniae*, *B. subtilis*, *B. cereus*, *E. coli*, *E. faecalis*, and *S. typhimurium* [15]. However, based on our findings, no inhibition was observed toward *E. coli* (Figure 6, Table 3).

Regarding the minimum inhibitory concentration (MIC), a value of 2.5 mg/mL was recorded against *S. aureus* ATCC 29213 and DSM 20231, while an MIC value of 5 mg/mL was determined for *S. aureus* ATCC 25923 and the three strains of *L. innocua*. MIC values > 10 mg/mL were recorded for *E. coli*.

In contrast, the *C. maritimum* hydroethanolic extract, even at the lowest concentration (0.25 mg/mL), strongly reduced the growth of every tested strain in the study by Souid et al. [15]. Hydroethanolic extracts from sea fennel flowers have an MIC value of 300 µg/mL against *S. aureus* ATCC 29213, according to Correia et al. [13]. This value is very similar to that of the extracts from Souid et al. (0.25 mg/mL).

Finally, a minimum bactericidal concentration (MBC) of 10 mg/mL was measured against *S. aureus* ATCC 29213 and DSM 20231, whereas values > 10 mg/mL were reported for all other strains (Table 3). It can be inferred that our results are 1.6-times higher than the MBC (>600 µg/mL) for *S. aureus* ATCC 29213 reported by Correia et al. [13] when compared to the results from the hydroethanolic extract of flowers. However, the data on these parameters cannot be directly compared with the results of this study, as the leaves, flowers, and stems of sea fennel are used separately in some cases, while the by-product contains these parts in varying proportions.

## 4. Conclusions

To valorize the previously unknown by-product of *C. maritimum* cultivation and minimize the environmental impact of its transformation, aqueous and hydroalcoholic extractions were performed. Promising results were obtained for both extracts. Both aqueous and hydroethanolic extracts could be used in food formulations to increase antioxidant capacity against lipid peroxidation, as assessed by the β-carotene bleaching test, although the hydroethanolic extract contained higher levels of phenolic acids and flavonoids. Further investigations could explore the application of these extracts in food systems to enhance preservation, especially when combined with other ingredients.

The aqueous extract was not effective for antimicrobial purposes against the eight pathogen strains tested, which included species such as *S. aureus*, *E. coli*, and *L. innocua*. In contrast, the hydroethanolic extract showed inhibition in the growth against *S. aureus* and *L. innocua* but not *E. coli*. However, the response of *S. aureus* and *L. innocua* strains to the extract varied. The minimum inhibitory concentration (MIC) and the minimum bactericidal concentration (MBC) varied among pathogenic microorganisms and their strains. Therefore, the optimization of these extracts could be a valuable strategy to provide the market with “clean label” preservatives. Future research could focus on evaluating the application of these extracts in real food systems to enhance preservation efficacy, particularly in combination with other natural preservatives or functional ingredients. Additionally, the in vivo potential activity of sea fennel has been emphasized for further exploration.

## Figures and Tables

**Figure 1 foods-14-02304-f001:**
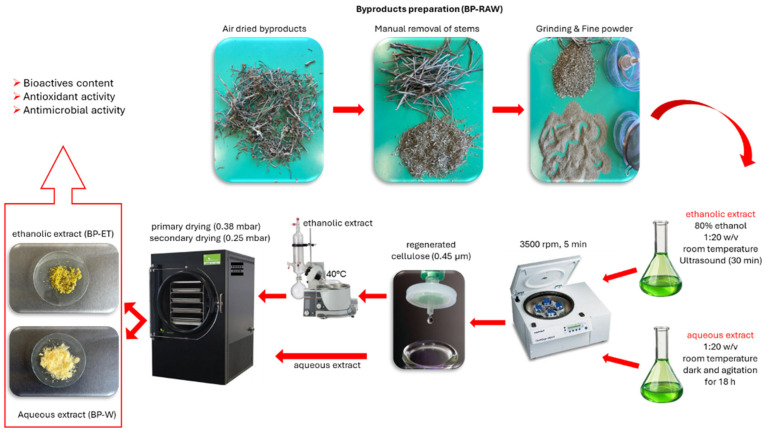
Sea fennel BP-RAW, BP-W and BP-ET preparation.

**Figure 2 foods-14-02304-f002:**
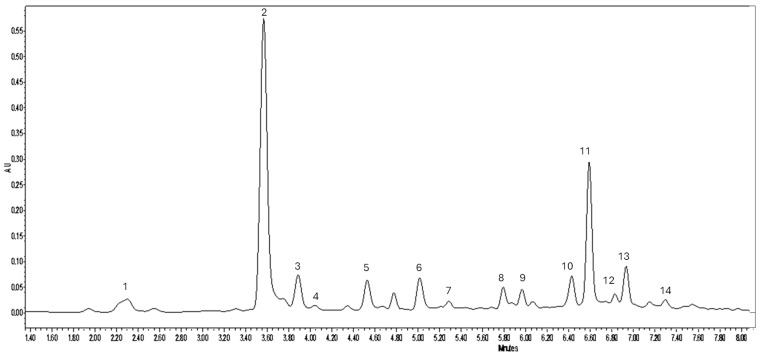
PDA chromatogram extracted at 280 nm of sea fennel hydroethanolic extract. Peaks 1–3: Caffeoylquinic acids (specifically peak 2 corresponds to chlorogenic acid); peak 4: caffeic acid; Peaks 5, 7: coumaroylquinic acids; peak 6: feruloylquinic acid; peak 8: quercetin-glucoside isomer; peak 9: isoquercetin; peaks 10, 11, 13: dicaffeoylquinic acids; peak 12: diosmin; peak 14: apigenin hexoside.

**Figure 3 foods-14-02304-f003:**
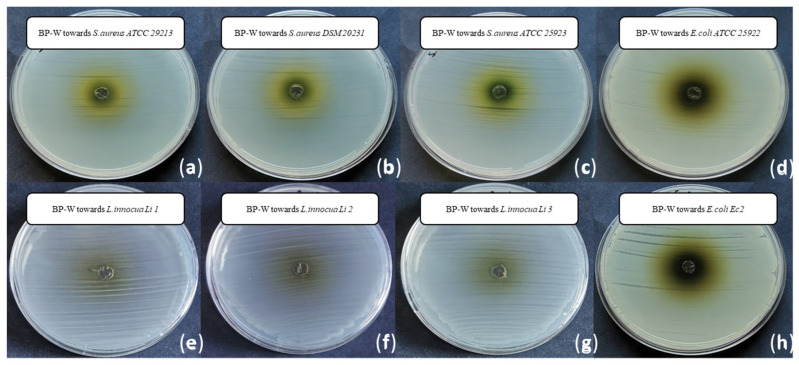
Inhibition growth zones determined through the agar well diffusion method, filling each well with 50 µL of the water extract from sea fennel by-product resuspended in water (400 mg mL^−1^) and sterilized by filtration, towards (**a**) *S. aureus* ATCC 29213, (**b**) *S. aureus* DSM 20231, (**c**) *S. aureus* ATCC 25923, (**d**) *E. coli* ATCC 25922, (**e**) *L. innocua Li* 1, (**f**) *L. innocua Li* 2, (**g**) *L. innocua Li* 3 and (**h**) *E. coli Ec* 2.

**Figure 4 foods-14-02304-f004:**
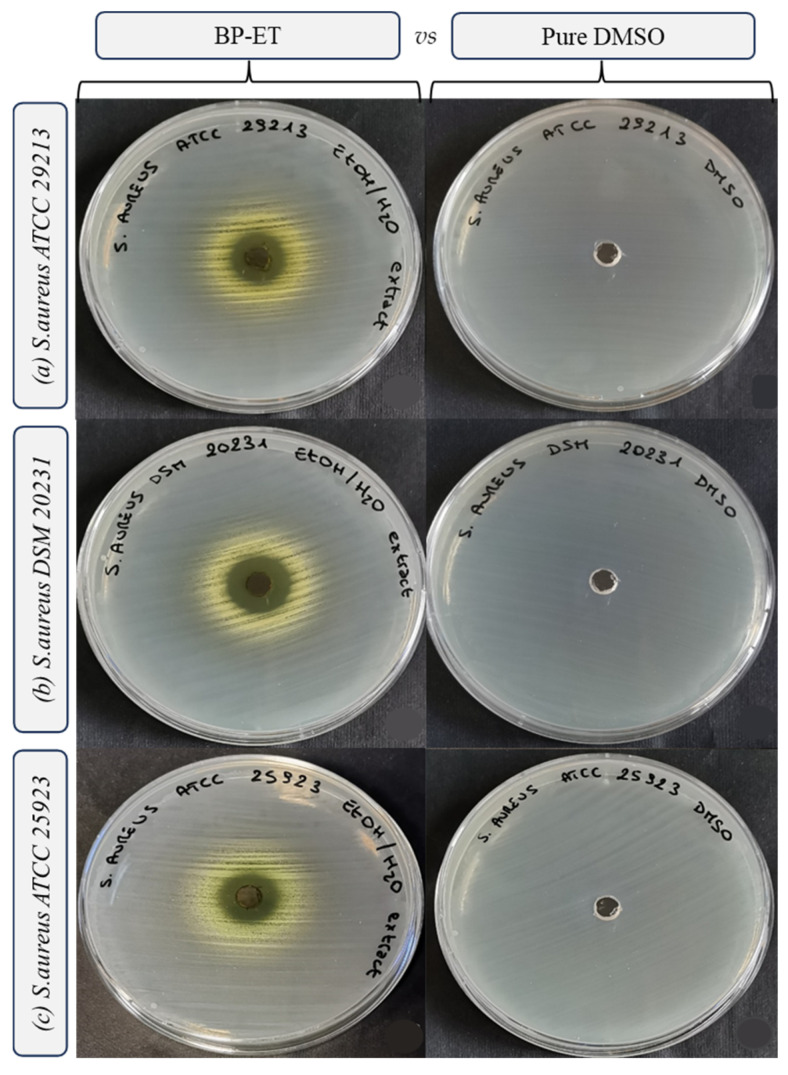
Inhibition growth zones determined through the agar well diffusion method, filling each well with 50 µL of ethanol extract from sea fennel by-product resuspended in DMSO (200 mg mL^−1^) vs. pure DMSO, towards *Staphylococcus aureus* (strain ATCC 29213 (**a**), DSM 20231 (**b**) and ATCC 25923 (**c**)).

**Figure 5 foods-14-02304-f005:**
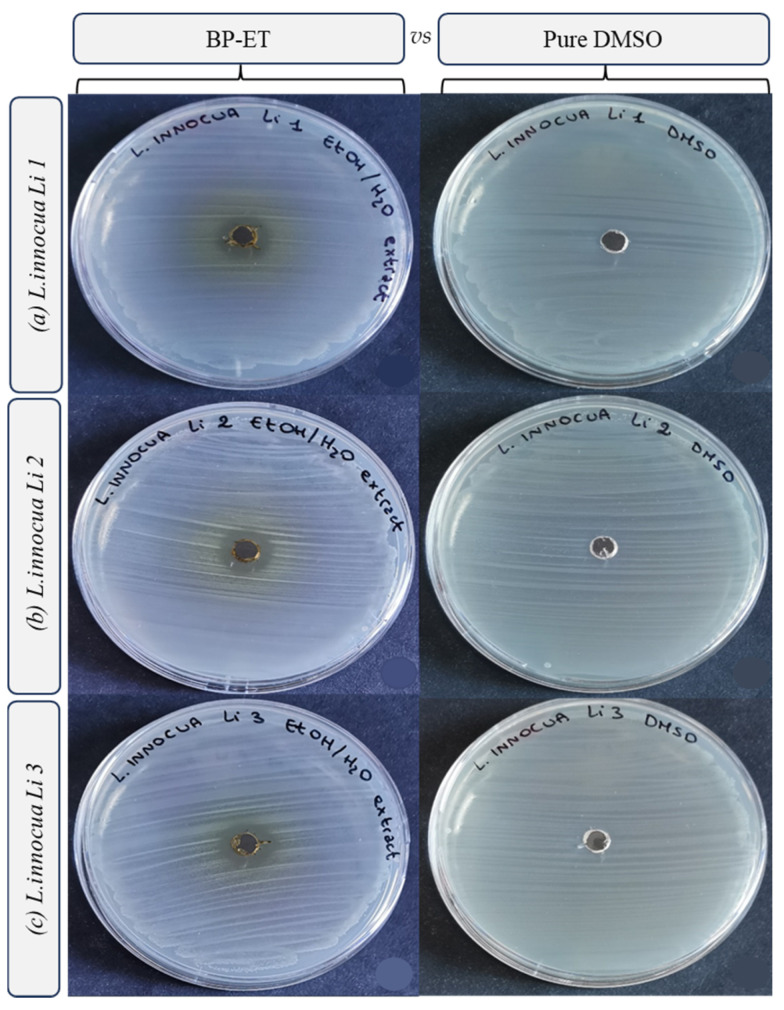
Inhibition growth zones determined through the agar well diffusion method, filling each well with 50 µL of ethanol extract from sea fennel by-product resuspended in DMSO (200 mg mL^−1^) vs. pure DMSO, towards *Listeria innocua* (strain Li 1 (**a**), Li 2 (**b**) and Li 3 (**c**)).

**Figure 6 foods-14-02304-f006:**
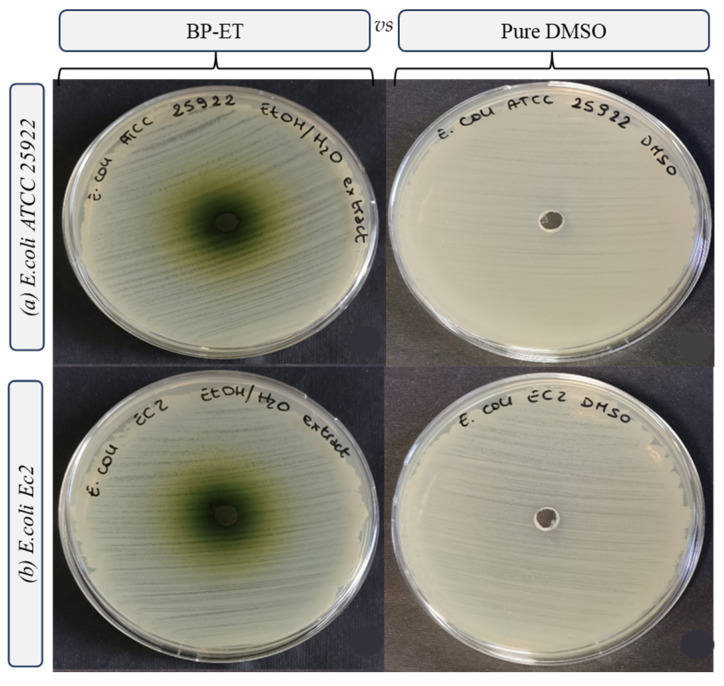
Inhibition growth zones determined through the agar well diffusion method, filling each well with 50 µL of ethanol extract from sea fennel by-product resuspended in DMSO (200 mg mL^−1^) vs. pure DMSO, towards *Escherichia coli* (strain ATCC 25922 (**a**) and Ec 2 (**b**)).

**Table 1 foods-14-02304-t001:** Polyphenol composition, TPC (total phenol content), TFC (total flavonoid content), and major carotenoids and tocopherols.

		BP-RAW	BP-W	BP-ET
	[M-H]-(*m*/*z*)	*Polyphenol Profile (mg CGA eq/g dw)*		
**Hydroxycinnamic acids and derivatives**		**4.31 ± 0.14**	**22.40 ± 2.22**	**22.99 ± 2.89**
Caffeoylquinic acid isomers (peaks 1–3)	353	2.77 ± 0.10 ^a^	14.16 ± 1.92 ^b^	13.55 ± 2.58 ^b^
Caffeic acid (peak 4)	179	0.09 ± 0.00 ^a^	0.30 ± 0.01 ^b^	0.31 ± 0.00 ^b^
Coumaroylquinic acids (peaks 5, 7)	337	0.45 ± 0.01 ^a^	1.63 ± 0.04 ^b^	1.52 ± 0.08 ^b^
Feruloylquinic acid (peak 6)	367	0.29 ± 0.02 ^a^	1.18 ± 0.07 ^b^	1.21 ± 0.02 ^b^
Dicaffeoylquinic acids (peaks 10, 11, 13)	515	0.70 ± 0.06 ^a^	5.13 ± 0.21 ^b^	6.40 ± 0.21 ^c^
*** Flavones and flavonols**		**0.71 ± 0.01**	**0.77 ± 0.03**	**1.69 ± 0.03**
Quercetin-glucoside isomer (peak 8)	475	0.20 ± 0.01 ^a^	0.24 ± 0.04 ^a^	0.57 ± 0.01 ^b^
Isoquercetin (peak 9)	463	0.18 ± 0.01 ^b^	0.13 ± 0.01 ^a^	0.55 ± 0.02 ^c^
Diosmin (peak 12)	607	0.24 ± 0.02 ^b^	0.10 ± 0.00 ^a^	0.27 ± 0.01 ^b^
Apigenin hexoside (peak 14)	431	0.09 ± 0.00 ^a^	0.30 ± 0.01 ^b^	0.30 ± 0.02 ^b^
**TPC** (mg GAE/g dw)		27.93 ± 1.4 ^a^	49.27 ± 1.4 ^b^	55.10 ± 0.24 ^b^
**TFC** (mg QE/g dw)		2.31 ± 0.52 ^a^	5.27 ± 0.71 ^b^	8.62 ± 1.12 ^c^
** *Carotenoids and Tocopherols (mg/kg dw)* **				
Lutein		29.62 ± 0.46 ^a^	n.d.	56.41 ± 1.72 ^b^
α-tocopherol		18.98 ± 0.76 ^a^	n.d.	30.89 ± 1.45 ^b^
γ-tocopherol		19.23 ± 0.81 ^b^	n.d.	16.74 ± 0.38 ^a^

Data are expressed as mean ± standard deviation (n = 3). Different letters in each row indicate statistical differences (*p* < 0.05). n.d.: not detected. Sea fennel by-product (BP-RAW) and its water (BP-W) and hydroethanolic (BP-ET) extracts. * tentative identification based on PDA and MS spectra.

**Table 2 foods-14-02304-t002:** Antioxidant activity of aqueous and hydroethanolic extracts of *Chrithmum maritimum* by-product.

Sample	ABTS Test	DPPH Test	β-Carotene Bleaching Test		FRAP Test
			30 min Incubation	60 min Incubation	
	IC_50_ (μg/mL)	IC_50_ (μg/mL)	IC_50_ (μg/mL)	IC_50_ (μg/mL)	μM Fe (II)/g
BHT	-	-	-	-	63.19 ± 2.22
BP-W	120.75 ± 8.83 ^a^	119.91 ± 6.54 ^a^	6.31 ± 0.94 ^b^	4.39 ± 0.67 ^a^	44.78 ± 2.35 ^b^
BP-ET	121.82 ± 9.04 ^a^	354.55 ± 10.31 ^b^	3.38 ± 0.35 ^a^	3.27 ± 0.29 ^a^	60.20 ± 3.98 ^a^
***Sign*.**	ns	**	**	ns	**
Positive Controls					
Ascorbic acid	1.85 ± 0.15	5.11 ± 0.78	-	-	-
Propyl gallate	-	-	1.1 ± 0.04	0.09 ± 0.02	-

Data are reported to mean ± Standard Deviation (SD) (n = 3). Ascorbic acid, propyl gallate, and BHT were used as positive controls in antioxidant tests. Differences within and between groups were evaluated by one-way ANOVA followed by Tukey’s multiple range test. Results followed by different letters in the same column are significantly different at ** *p* < 0.05.

**Table 3 foods-14-02304-t003:** Inhibition growth zones, minimum inhibitory concentration (MIC) and minimum bactericidal concentration (MBC) of the ethanol extract from sea fennel by-product on *Staphylococcus aureus* (strain ATCC 29213, DSM 20231 and ATCC 25923), *Listeria innocua* (strain Li 1, Li 2 and Li 3) and *Escherichia coli* (strain ATCC 25922 and Ec 2).

Species	Strain	Inhibition Growth Zones (mm)	MIC (mg mL^−1^)	MBC (mg mL^−1^)
*Staphylococcus aureus*	ATCC 29213	3.31 ± 0.09 ^b^	2.5 ± 0.0 ^c^	10.0 ^b^
*Staphylococcus aureus*	DSM 20231	4.50 ± 0.00 ^a^	2.5 ± 0.0 ^c^	10.0 ^b^
*Staphylococcus aureus*	ATCC 25923	3.13 ± 0.00 ^b^	5.0 ± 0.0 ^b^	>10.0 ^a^
*Listeria innocua*	Li 1	3.00 ± 0.18 ^b^	5.0 ± 0.0 ^b^	>10.0 ^a^
*Listeria innocua*	Li 2	3.13 ± 0.00 ^b^	5.0 ± 0.0 ^b^	>10.0 ^a^
*Listeria innocua*	Li 3	3.19 ± 0.27 ^b^	5.0 ± 0.0 ^b^	>10.0 ^a^
*Escherichia coli*	ATCC 25922	n.d. ^c^	>10.0 ^a^	>10.0 ^a^
*Escherichia coli*	Ec 2	n.d. ^c^	>10.0 ^a^	>10.0 ^a^

For each parameter (inhibition growth zone, MIC or MBC), values labelled with different letters in the same column are significantly different (*p* < 0.05). n.d. not detected.

## Data Availability

The original contributions presented in this study are included in the article. Further inquiries can be directed to the corresponding authors.

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
