# Peer review of "Food-Grade Polar Extracts from Sea Fennel (*Crithmum maritimum* L.) By-Products: Unlocking Potential for the Food Industry"

_foods, 2025, doi:10.3390/foods14132304_

Round 1

Reviewer 1 Report

Comments and Suggestions for Authors

Suggestions for Improvement

  1. English Language and Clarity

While the manuscript is understandable, there are numerous grammatical issues, overly long sentences, and awkward phrasings throughout.

Recommendation: A thorough language edit by a native speaker or professional editing service is highly recommended.

  1. Graphical Elements

Several figures (especially Figs. 3–6) are overly cluttered or lack detailed captions. It's difficult to distinguish extract effect from control (e.g., DMSO) visually in a few panels.

Recommendation: Improve clarity, resolution, and annotation of the images; ensure consistent formatting of all figures and tables.

  1. Novelty and Contribution

The novelty is clearly stated, but could benefit from a stronger comparative framing—e.g., how do the present MIC/MBC results compare in effectiveness with other natural preservatives or sea fennel extracts from non-by-product plant parts?

  1. Statistical Representation

Statistical significance indicators are inconsistently formatted (e.g., "p < 0.05" and "p< 0.05" are both used). Use standardized notation throughout.

Include R² values or correlation analysis plots for phenolic content versus antioxidant activity to visually support the positive correlations described.

  1. Discussion Section Depth

While the results are well presented, the discussion could be more critical, especially when comparing the antimicrobial efficacy to other literature.

For example, the smaller inhibition zones observed here could be due to the mixed tissue (by-product) origin—this should be acknowledged as a potential limiting factor.

  1. Minor Points

Consider revising long compound phrases like “antioxidant and antimicrobial capacity for food preservation purposes” to clearer alternatives, e.g., “for use as food-grade natural preservatives.”

Acronyms like TPC and TFC should be defined at first mention in both abstract and main text.

Table 1 and 3 could be split or simplified for better readability.

Comments on the Quality of English Language

While the manuscript is understandable, there are numerous grammatical issues, overly long sentences, and awkward phrasings throughout.

Recommendation: A thorough language edit by a native speaker or professional editing service is highly recommended.

Author Response

Suggestions for Improvement

  1. English Language and Clarity

While the manuscript is understandable, there are numerous grammatical issues, overly long sentences, and awkward phrasings throughout.

Recommendation: A thorough language edit by a native speaker or professional editing service is highly recommended.

Answer: We appreciate the reviewer’s concern. The manuscript has been read by a native English speaker and the quality of the language has been improved.

  1. Graphical Elements

Several figures (especially Figs. 3–6) are overly cluttered or lack detailed captions. It's difficult to distinguish extract effect from control (e.g., DMSO) visually in a few panels.

Recommendation: Improve clarity, resolution, and annotation of the images; ensure consistent formatting of all figures and tables.

Answer: Figures’ resolution, clarity and annotation have been improved.

  1. Novelty and Contribution

The novelty is clearly stated, but could benefit from a stronger comparative framing—e.g., how do the present MIC/MBC results compare in effectiveness with other natural preservatives or sea fennel extracts from non-by-product plant parts?

Answer: We thank the reviewer for this insightful comment. To strengthen the comparative context of our MIC/MBC findings, we have added more details in the Discussion section, enhancing how the antimicrobial performance of our hydroethanolic extract from sea fennel by-products (Lines: 484 to 502).

  1. Statistical Representation

Statistical significance indicators are inconsistently formatted (e.g., "p < 0.05" and "p< 0.05" are both used). Use standardized notation throughout.

Answer: Corrected throughout the manuscript.

Include R² values or correlation analysis plots for phenolic content versus antioxidant activity to visually support the positive correlations described.

Answer: Correlations have been calculated and added to the text (Lines: 406 to 408). 

Discussion Section Depth

While the results are well presented, the discussion could be more critical, especially when comparing the antimicrobial efficacy to other literature.

For example, the smaller inhibition zones observed here could be due to the mixed tissue (by-product) origin—this should be acknowledged as a potential limiting factor.

Answer: The authors appreciate the reviewer’s concern; we have acknowledged this fact and enhanced the discussion regarding antimicrobial efficacy. (Lines: 494 to 502).

  1. Minor Points

Consider revising long compound phrases like “antioxidant and antimicrobial capacity for food preservation purposes” to clearer alternatives, e.g., “for use as food-grade natural preservatives.”

Answer: Done

Acronyms like TPC and TFC should be defined at first mention in both abstract and main text.

Answer: Done

Table 1 and 3 could be split or simplified for better readability.

Answer: Done, table 1 and 3 have been simplified.

Comments on the Quality of English Language

While the manuscript is understandable, there are numerous grammatical issues, overly long sentences, and awkward phrasings throughout.

Recommendation: A thorough language edit by a native speaker or professional editing service is highly recommended.

Answer: We appreciate the reviewer’s concern. The manuscript has been read by a native English speaker and the quality of the language has been improved.

Reviewer 2 Report

Comments and Suggestions for Authors

The manuscript explores the valorization of sea fennel by-products using aqueous and hydroethanolic extractions. The following key issues should be addressed:  

  1. The by-product is a heterogeneous mixture of woody parts, stems, old leaves, and flowers. However, these components were not separated or analyzed individually. Given the likely variation in phenolic and flavonoid content among materials, why were the samples not fractionated? Furthermore, how is consistency in active compound content ensured in such a mixed material?  
  2. Only aqueous and 80% hydroethanolic extracts were tested, despite the manuscript acknowledging that other organic solvents may show stronger antimicrobial activity. While food-grade solvents are appropriate, some low-boiling solvents can be safely used after complete drying. Why was 80% ethanol chosen specifically—was this based on prior studies or optimization?  
  3. The extraction protocol lacks optimization. Key parameters such as solvent concentration, temperature, and extraction time were not explored. Without optimization, it is unclear whether the extraction yield is near optimal.  
  4. The identification of phenolic compounds via LC-MS is not sufficiently described. Were identifications based on standards, MS fragmentation patterns, or database matching? Detailed methodology is needed to assess the accuracy of compound identification.  
  5. Figure 2 raises several concerns. For example, peak 1 appears to represent overlapping compounds, yet it is grouped as a single peak. Additionally, peaks are labeled by broad compound classes (e.g., “Caffeoylquinic acids”) instead of specific chemical names. If individual compounds were identified, they should be explicitly labeled.

Author Response

Reviewer 2

The manuscript explores the valorization of sea fennel by-products using aqueous and hydroethanolic extractions. The following key issues should be addressed:  

  1. The by-product is a heterogeneous mixture of woody parts, stems, old leaves, and flowers. However, these components were not separated or analyzed individually. Given the likely variation in phenolic and flavonoid content among materials, why were the samples not fractionated? Furthermore, how is consistency in active compound content ensured in such a mixed material?  

Answer: We appreciate the reviewer’s insightful observation. As correctly noted, the by-product used in this study comprises a heterogeneous mix of plant tissues (woody stems, old leaves, and flowers), which were not analyzed separately. Our primary objective was to assess the feasibility of valorizing this by-product as it is generated from standard sea fennel cultivation and pruning practices. This approach reflects real conditions and aims to evaluate its potential application as a food-grade extract without extra cost or labor load for fractionation, thereby aligning with principles of circular economy and industrial scalability. We acknowledge that the phenolic and flavonoid contents likely vary among the individual plant parts. However, this studied matrix represents the actual composition of post-harvest residual biomass sourced from a single cultivation cycle and location, helping to reduce variability linked to environmental and agronomic factors, which they established to be causing variation in the bioactives composition of plant material. 

  1. Only aqueous and 80% hydroethanolic extracts were tested, despite the manuscript acknowledging that other organic solvents may show stronger antimicrobial activity. While food-grade solvents are appropriate, some low-boiling solvents can be safely used after complete drying. Why was 80% ethanol chosen specifically—was this based on prior studies or optimization?  

Answer: We thank the reviewer for the concern, our extraction strategy was intentionally designed to prioritize food-grade, GRAS (Generally Recognized As Safe), and environmentally sustainable solvents suitable for direct application in food systems. Our study presents preliminary results focused on evaluating the potential of sea fennel by-products as a source of bioactive compounds and their potential for further applications in the food industry. In future studies, we will certainly consider the findings of this work to explore and optimize alternative extraction strategies, including additional solvents or techniques that remain compliant with food safety standards.

  1. The extraction protocol lacks optimization. Key parameters such as solvent concentration, temperature, and extraction time were not explored. Without optimization, it is unclear whether the extraction yield is near optimal.  

As we mentioned, this study is exploratory for the potential use of sea fennel by-products and the results are preliminary for the future continuation of this work. In future studies, we will consider exploring and optimizing extraction strategies, including extraction time, additional solvents or techniques that remain compliant with food safety standards.

  1. The identification of phenolic compounds via LC-MS is not sufficiently described. Were identifications based on standards, MS fragmentation patterns, or database matching? Detailed methodology is needed to assess the accuracy of compound identification.  

Answer: We appreciate the reviewer’s comment, the tentative identification for the phenolic compounds was done based on retention time, absorbance spectra by PDA and mass-to-charge ratios (m/z) obtained in negative ionization mode. The detailed methodology for identification and semi-quantification of the phenolic compounds has been included in lines (166-172).  

  1. Figure 2 raises several concerns. For example, peak 1 appears to represent overlapping compounds, yet it is grouped as a single peak. Additionally, peaks are labeled by broad compound classes (e.g., “Caffeoylquinic acids”) instead of specific chemical names. If individual compounds were identified, they should be explicitly labeled.

Answer: We thank the reviewer for this important observation regarding Figure 2. Due to the complexity of the sample matrix and the resolution of the chromatographic conditions used, some peaks, particularly those corresponding to hydroxycinnamic acid derivatives, may indeed reflect overlapping isomers. As a result, we chose to label them under broader compound classes (e.g., “caffeoylquinic acids”) to avoid over-interpretation in the absence of confirmation. Where possible, specific compound names were included in the legend of Figure 2 (Peak 2, Peak 4, Peak 6, Peak 8, Peak 9, Peak 12 and Peak 14).
